# Unmasking Nuances Affecting Loneliness: Using Digital Behavioural Markers to Understand Social and Emotional Loneliness in College Students

**DOI:** 10.3390/s25061903

**Published:** 2025-03-19

**Authors:** Malik Muhammad Qirtas, Evi Zafeiridi, Eleanor Bantry White, Dirk Pesch

**Affiliations:** 1School of Computer Science and Information Technology, University College Cork, T12 K8AF Cork, Ireland; ezafeiridi@ucc.ie (E.Z.); dirk.pesch@ucc.ie (D.P.); 2School of Applied Social Studies, University College Cork, T12 K8AF Cork, Ireland; e.bantrywhite@ucc.ie

**Keywords:** digital phenotyping, loneliness mobile, mobile health, machine learning, passive sensing, wearables

## Abstract

Loneliness is a global issue which is particularly prevalent among college students, where it poses risks to mental health and academic success. Chronic loneliness can manifest in two primary forms: social loneliness, which is defined by a lack of belonging or a social network, and emotional loneliness, which comes from the absence of deep, meaningful connections. Differentiating between these forms is crucial for designing personalized and targeted interventions. Passive sensing technology offers a promising, unobtrusive approach to detecting loneliness by using behavioural data collected from smartphones and wearables. This study investigates behavioural patterns associated with social and emotional loneliness using passively sensed data from a student population. Our objectives were to (1) identify behavioural patterns linked to social and emotional loneliness, (2) evaluate the predictive power of these patterns for classifying loneliness types, and (3) determine the most significant digital markers used by machine learning models in loneliness prediction. Using statistical analysis, machine learning, and SHAP-based feature importance methods, we identified significant differences in behaviours between socially and Emotionally Lonely students. Specifically, there were distinct differences in phone use and location-based features. Our machine learning analysis shows a strong ability to classify types of loneliness accurately. The XGBoost model achieved the highest accuracy (78.48%) in predicting loneliness. Feature importance analysis found the critical role of phone usage and location-based features in distinguishing between social and emotional loneliness.

## 1. Introduction

Loneliness has become a widespread and severe problem in recent decades [1]. Perlman defines loneliness as a perceived discrepancy in the quality or quantity of one’s relationships [2]. While occasional loneliness is normal, its persistence can lead to serious physical, emotional and mental health issues, including higher blood pressure, increased cardiovascular disease risk and a rise in depressive symptoms [3,4,5]. Despite often being perceived as a singular state, loneliness is multifaceted and includes social and emotional dimensions [6]. Social loneliness refers to feeling disconnected due to a lack of desired social relationships, while emotional loneliness arises from the absence of deep, personal connections with others [7,8]. These dimensions do not always coincide; for instance, individuals with wide social networks may still feel Emotionally Lonely due to a lack of meaningful connections [9,10]. Given the distinct psychological and physiological implications of each type, differentiating them is crucial for effective interventions [11,12]. Traditionally, loneliness has been assessed using validated self-report scales such as the UCLA Loneliness Scale [13], but these methods rely on subjective recall and may not capture momentary or unconscious behavioural patterns. Existing studies have linked loneliness with reduced mobility, irregular sleep patterns and decreased social interactions [12]. While prior research has identified these distinct types, to the best of our knowledge, no study has investigated how they manifest in behavioural patterns derived from passively sensed data within the student population specifically.

College students represent a particularly salient group for the study of loneliness, given the unique set of challenges and transitions they encounter during this pivotal life stage. As they navigate the transition from adolescence to adulthood, students are often challenged by a new social environment, frequently far from their family support and their established social networks [14]. This challenge is coupled with the demands of academic stress, the pressure to conform to new social norms, and the challenge of establishing a sense of identity and belonging in an unfamiliar setting. Prior research has shed light on the prevalence of loneliness among this demographic; for instance, an extensive survey of around 33,000 college students found that a two-thirds majority had difficulties with loneliness and a sense of isolation [15]. This is alarming, considering that such levels of loneliness have been correlated with a range of negative mental health outcomes, including increased rates of depression, anxiety, and stress. Therefore, identifying behavioural markers of loneliness in the college student experience is a crucial step toward developing more effective interventions. These efforts not only hold the potential to improve student well-being but also have broader implications for public health and educational systems.

In response to the need to better understand and address loneliness among college students, the collection of behavioural data through passive sensing holds promise for detecting loneliness and contributing to a deeper understanding of this experience. Utilizing sensors embedded in ubiquitous devices such as smartphones and wearables such as a fitness tracker or smartwatch, passive sensing data collection is non-intrusive and continuous, thereby offering a comprehensive and objective snapshot of an individual’s daily life [16]. These data can include physical activity levels, sleep patterns, smartphone usage, and social interaction, as measured through call and message logs, and meeting others measured through Bluetooth device encounters. Unlike self-reported surveys and questionnaires, which can be subject to recall bias and social desirability, passively sensed data provide an unfiltered and unbiased lens into an individual’s behaviour and well-being. In the context of loneliness, this technology has the potential to uncover subtle but telling signs of both social and emotional loneliness. Therefore, passive sensing stands out as an excellent tool, not only for detecting loneliness in a timely and precise manner but also for shedding light on the nuanced ways in which loneliness manifests in daily life.

Advances in digital technology now allow passive sensing to infer loneliness by continuously tracking behaviours via smartphones, wearables and ambient sensors. These methods provide an objective alternative to self-reported surveys, which are prone to recall bias. A systematic review of studies highlights the increasing reliance on passive sensing approaches, with 69% of studies using smartphone and wearable devices to infer behavioural markers associated with loneliness [17]. These methods analyse mobility patterns, communication logs (e.g., call and SMS activity) and in-home behaviours to identify individuals experiencing loneliness. Recent work has also explored personalized models for loneliness detection, where grouping individuals based on behavioural similarities improves predictive accuracy [18]. Additionally, multi-device sensing approaches that combine data from smartphones, smartwatches and other wearable devices have shown improved loneliness detection by integrating physiological and behavioural signals [19]. These studies highlight the growing reliance on passive sensing methodologies to overcome the limitations of traditional self-report measures. However, while passive sensing has been used for general loneliness detection, its ability to differentiate between social and emotional loneliness in students remains unexplored.

Research on social and emotional loneliness has mostly used statistical models to analyse psychological and demographic factors. Studies on elderly populations have shown that social loneliness is more prevalent in cognitively impaired individuals, while emotional loneliness remains relatively stable across dementia and non-dementia groups [20]. Large-scale community studies have also linked social loneliness to reduced social engagement and lower activity levels, whereas emotional loneliness is more closely associated with widowhood and lack of emotional support [21,22]. However, current approaches lack fine-grained differentiation between social and emotional loneliness in younger populations, specifically through a behavioural lens.

The objective of this study is to investigate social and emotional loneliness using passively sensed data for a student population. Our goal is to determine the subtle differences in data from individuals who are socially and Emotionally Lonely, evaluate the predictive power of behavioural features in categorizing these loneliness types and find the behavioural features that are most important for differentiating between these two types of loneliness. The key research questions are as follows.

Can behavioural features extracted from passively sensed data differentiate between socially and Emotionally Lonely students?Can behavioural features help to classify social and emotional loneliness?What behavioural features are most important for predictive models in classifying loneliness and its types?

## 2. Methods

### 2.1. Dataset

We used an existing dataset of passively sensed data by the University of Washington. The data were collected during the Spring quarter of 2019 over 10 weeks from March to June [23]. Data were collected from 218 undergraduate students who participated via email and social media invitations. Detailed demographic information about the participants is provided in Table 1.

The AWARE smartphone application was used for data collection, which works in the background without requiring any user interaction [24]. Additionally, Fitbit wearable fitness trackers were used to collect data on sleep and physical activity. The dataset includes multiple sensors, which include Bluetooth, Wi-Fi, location, phone usage, call activity, physical activity, and sleep patterns. The study was approved by the University of Washington’s Institutional Review Board (IRB number: STUDY00003244), and all participants provided informed consent. Data confidentiality was ensured through strict adherence to anonymization protocols, with access to identifiable information restricted to the core data team. Additionally, data from participants who chose to withdraw from the study were promptly removed from the dataset. The detailed flow of our methods section has been provided in Figure 1.

### 2.2. Data Preprocessing

We included only those students who completed the post-study loneliness questionnaire, leading to 205 students out of 218. The reason for selecting post-study completion was to assess the effects of loneliness throughout the study period, which would provide a comprehensive view of each participant’s experience. We then identified and removed outliers using the z-score method. To handle the issue of missing values, we filled in missing data for each student using the median value for continuous data features specific to a session. For categorical data, we used the mode value of a particular feature for that session. Categorical features underwent one-hot encoding to achieve integer representation. Numerical data were normalized using min-max scaling, which adjusted each value to a range between 0 and 1.

The dataset extracted behavioural features using the Reproducible Analysis Pipeline for Data Streams (RAPIDS) [25]. The dataset provided day-level and segmented daily interval features spanning from 12:00 a.m. to 11:59 p.m. These intervals were further subdivided into distinct time segments: morning (6 a.m.–12 p.m.), afternoon (12 p.m.–6 p.m.), evening (6 p.m.–12 a.m.), and night (12 a.m.–6 a.m.). The division into specific time segments serves an important role in capturing nuanced behavioural patterns, as individuals tend to engage in distinct activities during different parts of the day. These patterns, like routines and variability, were measured using metrics such as counts, standard deviations, and entropy. A detailed overview of these features is provided in RAPIDS documentation [25,26]. A total of 403 features were extracted for each participant. All sensor features were extracted for 5 time segments (day, morning, afternoon, evening, and night) except for sleep features. After preprocessing, the dataset consists of 14,350 samples, with each sample representing one day of data for a participant. For a summarized form of the extracted features, please refer to Table 2. A sample of the feature matrix scheme is provided in Figure 2.

### 2.3. Categorizing Social and Emotional Loneliness

In the dataset, loneliness was assessed using a revised 10-item UCLA Loneliness Scale [13]. The revised 10-item UCLA Loneliness Scale uses a 4-point response scale for each item, where 1 represents ‘never’, 2 ‘rarely’, 3 ‘sometimes’, and 4 ‘often’. With 10 items in total, the minimum possible score is 10 (if a participant responds with 1 for all 10 items), and the maximum possible score is 40 (if a participant responds with 4 for all 10 items). Therefore, the overall score range for this 10-item questionnaire spans from 10 to 40. For our research purposes, we divided the items of the scale into two distinct categories of social and emotional loneliness based on the criteria proposed in [27,28]. Items in the scale that are related to feeling isolated, feeling left out, and lack of companionship and social interactions are classified under the social loneliness category, while the items related to emotional disconnect, like not feeling close to others and not being truly understood by anyone, are put under the emotional loneliness category. Each category consisted of 5 items from the original scale presented in Table 3. Some items in the scale, denoted by (R), are reverse-scored. This means that for these items, the scoring is inverted: a response of 1 is scored as 4, 2 as 3, 3 as 2, and 4 as 1. This reverse scoring ensures that higher scores consistently indicate greater levels of loneliness across all items.

There is no universally accepted threshold in the literature for determining loneliness cut-off scores to divide into low or high loneliness categories. Many studies have proposed their own cutoff scores, such as one by Cacioppo et al. [4]. In our study, we consider scoring in two dimensions: social loneliness (referred to as social_score) and emotional loneliness (referred to as emotional_score), each ranging from 5 to 20. A cumulative score of 10 is achieved if a participant selected ‘rarely’ for all 5 questions, which represents occasional feelings of social or emotional loneliness. Hence, we chose 10 as our cut-off score for each subscale. To categorize loneliness levels, we used the following approach:Participants with a social_score of above 10 and an emotional_score of 10 and below were labelled as ‘Socially Lonely’.Participants with an emotional_score of above 10 and social_score of 10 and below were labelled as ‘Emotionally Lonely’.Participants scoring above 10 on both scales were considered ‘both socially and Emotionally Lonely’.Finally, those scoring 10 or below on both scales were categorized as ‘not lonely’.

We first applied some basic statistics to analyse the different aspects of loneliness among the students in the dataset. This involved calculating the mean, median, first quartile (Q1), third quartile (Q3), and standard deviation (SD) for the four distinct categories: ‘Socially Lonely’, Émotionally Lonely’, ‘both socially and Emotionally Lonely’, and ‘not lonely’. This provided an initial overview of the overall distribution and central tendency of loneliness within the dataset.

### 2.4. Differentiating Social and Emotional Loneliness

We conducted a statistical analysis using behavioural features to address our first research question about differentiating between social and emotional loneliness. Before applying statistical analysis, we used Mutual Information (MI) for feature selection [29] to identify the most relevant behavioural features for differentiating between social and emotional loneliness. We selected features whose cumulative MI reached a threshold of 95% of the total MI across all features.

We then checked the normality of the distribution of behavioural features for both socially and Emotionally Lonely student groups. This was conducted using the Shapiro–Wilk test [30], which is a robust method for testing normality in data and is particularly effective for small sample sizes like ours (24 participants in the Socially Lonely group and 19 in the Emotionally Lonely group). The normality check is important because different statistical tests are suitable for different types of data distributions. Parametric tests, such as t-tests or ANOVA, assume that the data follow a normal distribution. If this assumption is violated, non-parametric tests, including the Mann–Whitney U test, are more appropriate because they do not rely on the normality assumption. By using the Shapiro–Wilk test, we ensured the validity and reliability of our subsequent statistical analyses in identifying differences between socially and Emotionally Lonely groups.

Given that the Shapiro–Wilk test indicated a non-normal distribution for majority features, we used a two-sided Mann–Whitney U test to compare the feature distributions between the socially and Emotionally Lonely groups. This non-parametric test was selected for its ability to compare differences between two independent samples without the assumption of normal distribution. The goal was to determine if there were statistically significant differences in the feature profiles between Socially Lonely and Emotionally Lonely groups. The null hypothesis for this test was that there was no difference in the mean of features between the two groups, while the alternative hypothesis stated that the mean of one or more features would differ.

To quantify the magnitude of the differences observed between the socially and Emotionally Lonely groups, we calculated effect sizes using the statistical technique called bootstrapping. Bootstrapping is a resampling technique that enhances the reliability of statistical inferences, which is particularly useful when working with smaller sample sizes. This method works by repeatedly sampling with replacements from the original dataset to create multiple simulated datasets. This helps with the estimation of sampling distributions and the calculation of robust statistics. In our case, a total of 10K bootstrap samples were used to estimate the distribution of the effect size and resampled features with replacements, which provides a more robust measure in the context of our small sample sizes. This involved resampling the observations (with replacement) 10K times to create multiple bootstrap samples. We selected Cohen’s *d* as our effect size metric. Cohen’s *d* is a standardized measure of the difference between two group means, which is expressed in units of standard deviations [31]. This standardization is used for meaningful comparisons across different variables with varying scales, which makes it suitable for our analysis of diverse features. Alongside the point estimate, a 95% confidence interval was computed from the bootstrap distribution to calculate the precision of the effect size.

### 2.5. Predictive Modelling for Loneliness Classification

We trained multiple ML multi-class classification algorithms to address our second research question regarding the power of behavioural features in classifying loneliness types. The dataset has instances labelled as ‘Socially Lonely’, ‘Emotionally Lonely’, ‘both lonely’, or ‘not lonely’. We selected four widely used machine learning algorithms for our classification task: Support Vector Machine (SVM) [32], XGBoost [33], Random Forest [34], and K-Nearest Neighbour (KNN) [35]. We chose these ML algorithms because they are widely used in supervised classification, easy to train, and interpretable [36]. Moreover, these ML algorithms have also been used in similar previous studies [37,38,39].

We used nested cross-validation for model evaluation and selection. In the outer loop, we used leave-one-subject-out cross-validation to assess generalization. Within each iteration of the outer loop, we used an inner loop of stratified three-fold cross-validation for hyperparameter tuning using GridSearchCV. Preprocessing steps were conducted within each inner fold before model training and evaluation, including feature scaling, handling missing values, feature selection using MI, and handling class imbalance with SMOTE. We used the macro-averaged F1 score as the evaluation metric for model selection in cross-validation.

For baseline comparisons, we considered the majority class, random weighted classifier, and decision tree models trained on the original, imbalanced dataset to provide a fair comparison. The majority class model always predicts the most frequent class in the dataset to serve as a naive baseline. The random weighted classifier makes predictions randomly. The decision tree model makes classifications based on a series of feature-based questions to serve as a middle-ground comparison between the simplest baselines and our more complex models. These baselines serve as reference points to evaluate the performance of more complex models. The primary metrics to assess the performance of each model were accuracy, precision, sensitivity, and F1 score. These metrics, which provide a detailed view of the model’s performance, were calculated for each loneliness category. Accuracy provided a global view of model performance, while precision, sensitivity, and the F1 score provided insights into the model’s ability to predict each specific class.

### 2.6. Feature Importance Analysis

To address our third research question regarding determining the most important features used by the predictive models in distinguishing between the different loneliness types, we used the Shapley Additive exPlanations (SHAP) values for XGBoost and Random Forest models [40]. We chose these two models specifically because they provided the best results in our predictive classification task.

SHAP values measure the impact of each feature on the model’s output for classification. In our multi-class context, SHAP computes values for each class separately to compute feature importance for each loneliness type. Once the classification models were trained, we used the SHAP library to compute the SHAP values for each feature across all data points. This process outputs a SHAP value for each feature for each prediction and each class, indicating the feature’s impact on the model’s output for classification. We averaged the absolute SHAP values for each feature within each class to analyse the importance of class-specific features. Averaging the absolute SHAP values helped us to see the overall influence of each feature on the model’s predictions to provide a clear and robust measure of feature importance within each class. We then ranked the features based on their average absolute SHAP values within each class to identify the most important behavioural indicators for each loneliness category.

## 3. Results

### 3.1. Overview of Loneliness in Participants

For the specific dimensions of loneliness, the mean social loneliness score was 10.93, with a median of 11 and an interquartile range (Q1: 9, Q3: 13) with a standard deviation of 2.736. Emotional loneliness scores have a mean of 10.71, a median of 11, and the same interquartile range (Q1: 9, Q3: 13) but a slightly higher standard deviation of 2.905. These statistics include all students to capture the full range of social and emotional loneliness within the dataset.

When classifying participants based on their social and emotional loneliness scores, we found a slightly larger subset experiencing social loneliness (11.71%, 24 out of 205) compared to emotional loneliness (9.27%, 19 out of 205). However, a significant portion (42.44%, 87 out of 205) reported feeling both socially and Emotionally Lonely, which highlights the interconnectedness of these experiences. Furthermore, 36.59% (75 out of 205) were classified as feeling neither socially nor Emotionally Lonely.

### 3.2. Statistical Differences for Social and Emotional Loneliness

To determine whether behavioural features could differentiate between social and emotional loneliness, we conducted a two-sided Mann–Whitney U test on the non-normally distributed data (as confirmed by the Shapiro–Wilk test). Table 4 presents the statistically significant (*p* < 0.05) features only with mean differences and their effect sizes for socially and Emotionally Lonely groups. We interpret the magnitude of effect sizes (Cohen’s *d*) using the commonly used approach, where values of 0.2, 0.5, and 0.8 are considered as thresholds for small, medium, and large effects, respectively [31]. Below are the key results.

Our analysis of location-based features found significant differences between Socially Lonely (SL) and Emotionally Lonely (EL) groups. The log-transformed location variance in the evening, which represents the variability in a participant’s geographic position, was lower for the SL group (M = 2.301 log units) compared to the EL group (M = 3.751 log units), with a medium effect size of −0.715. This indicates that Emotionally Lonely individuals show more varied movement patterns in the evening. On a daily basis, the SL group visited fewer significant places, defined as distinct locations (M = 1.504 places), compared to the EL group (M = 2.167 places). The number of transitions between these significant locations was also lower for the SL group (M = 5.463 transitions) than the EL group (M = 7.374 transitions), with a large effect size of −0.780, indicating less movement between key locations for Socially Lonely individuals. Interestingly, the normalized location entropy, which measures the evenness of time distribution across significant locations, was higher for the SL group (M = 0.451 entropy units) compared to the EL group (M = 0.323 entropy units). This indicates that while Socially Lonely individuals visit fewer locations, they tend to distribute their time more evenly across these places.

The analysis of phone usage patterns also found significant differences between the two groups. The EL group showed higher overall phone engagement with a total daily unlock duration of 495.535 min compared to 400.204 min for the SL group (d = −0.535). The EL group took longer to first use their phone after waking, averaging 45.067 min versus 28.745 min for the SL group with an effect size of −0.578. Another difference was in the maximum duration of a single unlock episode, with the EL group spending up to 18.073 min compared to 7.683 min for the SL group. Other metrics, including the frequency of phone unlocks, also showed higher values for the EL group.

The analysis of Bluetooth-related patterns also found significant differences between groups. It shows that the EL group encountered more unique devices daily (M = 5.516 encounters, 95) compared to the SL group (M = 3.701 encounters), which indicates increased social proximity or time spent in populated areas. Step count analysis showed the EL group had higher average daily steps (M = 5300.745 steps) than the SL group (M = 4800.335 steps), with a medium to large effect size (d = −0.754). Sleep patterns differed as well, with the SL group sleeping longer on average (M = 510.047 min) compared to the EL group (M = 407.731 min). However, the EL group spent more time awake in bed (M = 88.385 min) than the SL group (M = 60.320 min), which might indicate sleep quality issues.

### 3.3. Predictive Power of Behavioural Features in Loneliness Categories

Table 5 presents the predictive performance of different ML classifiers for categorizing loneliness into ‘Socially Lonely’, ‘Emotionally Lonely’, ‘both lonely’, and ‘not lonely’. We compared these classifiers against three baseline models: Majority Class (BL1: MC), Decision Tree (BL2: DT), and Random Weighted Classifier (BL3: RWC). These baseline models were chosen because they provide simple, interpretable benchmarks to evaluate the performance of more complex classifiers.

The XGBoost model achieved the highest overall accuracy of 78.48%. It outperformed in classifying the ‘Both Lonely’ category with an F1-score of 85.44% and showed strong performance in the ‘Emotionally Lonely’ category with an F1-score of 70.49%. The XGBoost model showed the best overall balance and highest metrics across all classes. The confusion matrix for XGboost is presented in Figure 3. After that, the Random Forest model achieved an accuracy of 75.58%, showing strong performance with an F1-score of 82.41% for the ‘Both Lonely’ class and 72.76% for the ‘Not Lonely’ class. The model provided a good balance of precision and sensitivity across all categories. Support Vector Machine (SVM) performed well with 70.10% accuracy and made strong predictions in the ‘Both Lonely’ and ‘Not Lonely’ categories. K-Nearest Neighbour (KNN) showed balanced performance with a 65.53% accuracy.

While F1-scores provide a balanced performance measure, investigating precision and sensitivity separately can provide important observations regarding these models. For instance, the XGBoost model showed high precision (88.07%) in detecting students who are ’Both Lonely’, which shows a low false positive rate for this category. However, its sensitivity for ’Socially Lonely’ students was lower (58.59%), which shows some challenges in identifying all cases in this category. The Random Forest model showed a more balanced precision–sensitivity trade-off for the ’Not Lonely’ category (precision: 75.94%, sensitivity: 70.86%), which shows consistent performance in correctly both identifying and capturing instances of this class. These nuances in precision and sensitivity across different loneliness categories highlight the varying challenges in accurately classifying each type of loneliness.

### 3.4. Important Features for Loneliness Classification

Figure 4 and Figure 5 present the mean absolute SHAP values for different features across the four loneliness categories: Socially Lonely (SL), Emotionally Lonely (EL), Both Lonely (BL), and Not Lonely (NL). The *x*-axis represents the normalized mean absolute SHAP value for each feature, which shows the average magnitude of that feature’s contribution to the model’s output. SHAP values represent the impact of a feature on the model’s prediction. The magnitude of a SHAP value shows the feature’s importance for a particular prediction. In our figures, larger mean absolute SHAP values indicate a stronger average influence of that feature on the model’s decisions.

Both the XGBoost and Random Forest models identified location-based and phone usage features as highly influential in distinguishing between loneliness categories. However, there were some differences in the relative importance of specific features between the two models. The most influential features in the XGBoost model were the maximum duration of phone usage and maximum length of stay at a location in clusters. These features showed a strong impact across all loneliness categories, but the strongest effect was on the ‘Both Lonely’ category. Sleep-related features such as average duration awake also showed some importance, specifically for the ‘Not Lonely’ category.

In our Random Forest model, the SHAP value analysis also found important findings related to features. Location-based features such as variance and entropy were highly influential and had the greatest impact on almost all categories. The maximum durations of the phone and location entropy were also influential. Interestingly, Bluetooth-related features appeared more important in the Random Forest model as compared to XGBoost.

## 4. Discussion

The main aim of this study was to explore whether behavioural features from passively sensed data could distinguish between social and emotional loneliness and classify the types of loneliness using machine learning models. The analysis found statistically significant differences in the behavioural markers for the social and emotional loneliness groups. The observed differences in behavioural features between the Socially Lonely (SL) and Emotionally Lonely (EL) groups provide evidence that passive sensing can capture distinct patterns associated with these two forms of loneliness.

Location-based behaviours showed significant differences between the two groups. The EL group showed higher location variance, especially in the evening, visited more significant places, and had more location transitions compared to the SL group. In contrast, the SL group’s lower mobility and the fewer significant places visited may indicate a lack of interest or opportunity to participate in social interactions, which is a sign of social loneliness [41]. This behaviour might be linked to the introversion–extraversion dimension of personality. Socially Lonely individuals may lean towards introversion and prefer solitary activities and less social stimulation [42]. There could also be some other factors for this behaviour, such as a lack of social support, which could discourage them from exploring new environments or engaging in social activities. Interestingly, while the SL group visited fewer locations, they showed higher location entropy, which indicates more evenly distributed time across the places they visited. This might indicate a preference for familiar or comfortable environments, maybe somehow to manage feelings of social disconnection. This difference between the two groups shows the nuanced ways in which loneliness manifests. However, it is important to note that the current data cannot definitively establish a causal relationship. While other studies provide further information about the reasons behind loneliness-related behaviours [43,44,45,46], a deeper investigation is still needed while using different methods like combining qualitative methods with behavioural features to gain a more nuanced understanding of the complex relationship between these two loneliness types and daily life behaviours.

The findings from phone usage patterns also show differences between the two types of loneliness. The EL group uses their phones more often and for longer periods, which may show their preference for digital communication instead of face-to-face interactions. This could be explained by the idea of the displacement hypothesis, where increased digital engagement takes the place of in-person social interactions and could lead to increased feelings of isolation or loneliness [47]. The SL group uses their phones less, maybe because they have a different social situation or do not feel the need for digital contact.

The Bluetooth data show more unique device encounters for the EL group, which supports the location data in suggesting that Emotionally Lonely individuals might seek out more populated areas. According to existing research, this behaviour aligns with the emotional loneliness construct, where individuals might feel lonely despite being in social settings [48]. Significant differences in physical activity and sleep patterns between the two groups provide additional information about the behavioural impacts of loneliness. The SL group is less physically active, which might cause or be a result of their withdrawing socially. They also sleep more, which could be due to many reasons, like if they are feeling depressed or not active during the day, leading to a need for less sleep at night.

These findings have important implications for understanding the complex nature of loneliness and its impact on student behaviour and well-being. These findings also show that passive sensing can differentiate between social and emotional loneliness and find the behavioural patterns associated with each. The findings can be used for targeted interventions, such as promoting social engagement for Socially Lonely students or addressing emotional needs and connectivity for Emotionally Lonely students. To maximize the effectiveness of such interventions, it is important to first gain a deeper understanding of the motivations and experiences behind the behavioural differences that we observed. Qualitative research could explore these behaviours to help us understand why individuals engage in specific patterns and how they perceive the connection between their behaviours and feelings of loneliness.

The machine learning models trained for classification based on loneliness types showed their predictive capabilities of behavioural features extracted from passively sensed data. The best-performing model was XGBoost, with an overall accuracy of 78.48%. This highlights the power of ensemble-based techniques in capturing complex patterns and non-linear relationships within sensor-derived features. Moreover, the F1 score for XGBoost is also high for the ‘Both Lonely’ category compared to other models. This shows the overall robustness of XGBoost for the classification of social and emotional loneliness.

The varying precision and sensitivity scores across different models and loneliness categories highlight important observations about the challenges of classifying loneliness types based on behavioural data. For instance, XGBoost’s high precision (88.07%) for the ‘Both Lonely’ category indicates that when the model predicts this category, it is highly likely to be correct. This indicates that students experiencing both social and emotional loneliness may exhibit more distinct behavioural patterns that are easier for the model to identify. On the other hand, the lower sensitivity for the ‘Socially Lonely’ category across models indicates that behaviours associated with social loneliness might be more subtle or varied, making it challenging for models to identify all instances. This could imply that social loneliness manifests in more diverse ways in behaviours and might overlap with patterns seen in other loneliness categories, which makes it difficult for models to identify it correctly.

The feature importance analysis for the XGBoost and Random Forest models provides important findings about the behavioural features that distinguish between different types of loneliness. The consistent importance of location-based and phone usage features across both models highlights their significance in understanding and categorizing loneliness. This indicates that mobility patterns play a significant role in differentiating loneliness types. Interestingly, our analysis found lower SHAP scores for features distinguishing between SL and EL groups compared to those differentiating the ‘Both Lonely’ and ‘Not Lonely’ categories. There could be different reasons for this observation. SL and EL might share similar behavioural patterns in some aspects, which made it challenging for the models to distinguish between them based solely on passive sensing data. This means the behavioural manifestations of social versus emotional loneliness might be more nuanced and less pronounced in passive sensing data compared to the clear differences between being lonely (in any form) and not lonely. This could be because students experiencing both types of loneliness have shown clearer or more consistent behaviours. At the same time, those who are not lonely have significantly different patterns of movement, phone usage, and sleep.

## 5. Conclusions

In this study, we explored the multifaceted nature of loneliness among students, specifically focusing on the distinction between social and emotional loneliness using passively sensed data. Our research has addressed three key questions that provide important insights into the behavioural manifestations of different types of loneliness. Our analysis found that behavioural features extracted through passively sensed data can differentiate between Socially and Emotionally Lonely students. The statistical test found statistically significant differences in various behavioural features between these two groups. Socially Lonely individuals showed less variance in their locations as compared to Emotionally Lonely individuals. Additionally, Socially Lonely students had shorter overall phone usage duration, fewer Bluetooth scans, and fewer steps than Emotionally Lonely students. We also showed the considerable predictive power of behavioural features in classifying social and emotional loneliness. The XGBoost model achieved the highest overall accuracy of 78.48% and a high F1 score across all types of loneliness. This shows the potential of behavioural features extracted through passive sensing in identifying and differentiating between types of loneliness. We identified the most important behavioural features for predictive models in classifying loneliness and its types. Our analysis identified phone usage and location-based features as critical in distinguishing between loneliness categories in both XGBoost and Random Forest models. The findings highlighted that phone usage duration, location variance, and sleep-related features were particularly significant in differentiating Socially, Emotionally, and both Socially and Emotionally Lonely individuals from those not experiencing loneliness. Despite these significant findings, our study has some limitations that could be addressed in future research. The generalizability of our findings is limited by the size and diversity of the dataset. The participants, who are college students, form a homogeneous group with possibly similar daily routines and psychological challenges. The dataset that we used does not capture all the characteristics associated with loneliness. Personal relationships [49], age [50], major life events [51], and mental health history [52] all significantly influence an individual’s experience of loneliness. Future research should aim to include more diverse populations with richer demographic data to broaden the applicability of these findings.

## Figures and Tables

**Figure 1 sensors-25-01903-f001:**
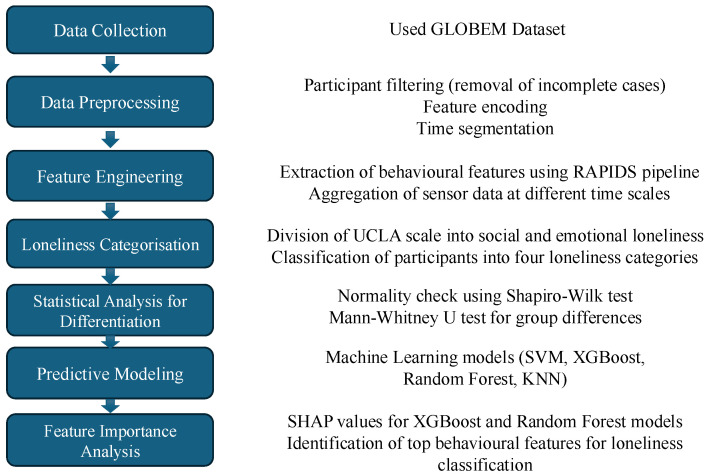
Methodology flowchart for loneliness classification using behavioural data from the GLOBEM dataset.

**Figure 2 sensors-25-01903-f002:**
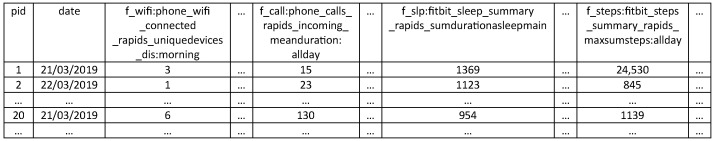
Sample schema of the feature matrix (each column is a feature and each row is a sample per participant).

**Figure 3 sensors-25-01903-f003:**
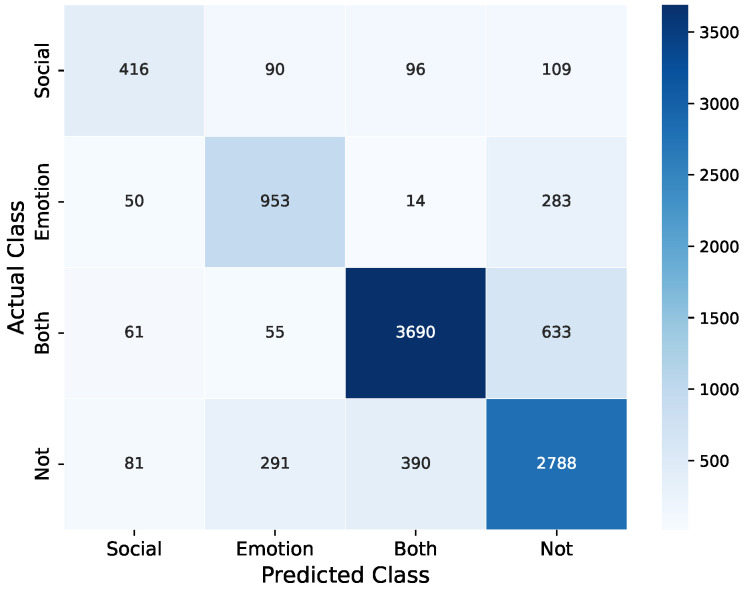
Confusion matrix for the XGBoost model in classifying social and emotional loneliness.

**Figure 4 sensors-25-01903-f004:**
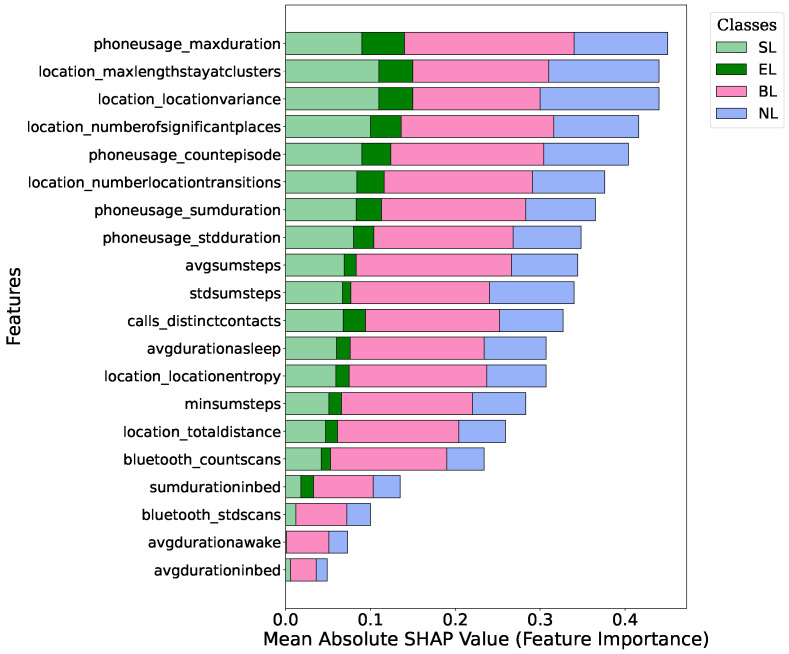
Mean absolute SHAP values representing feature importance in the XGBoost model. Higher values indicate stronger contributions of a feature to the model’s decision-making process.

**Figure 5 sensors-25-01903-f005:**
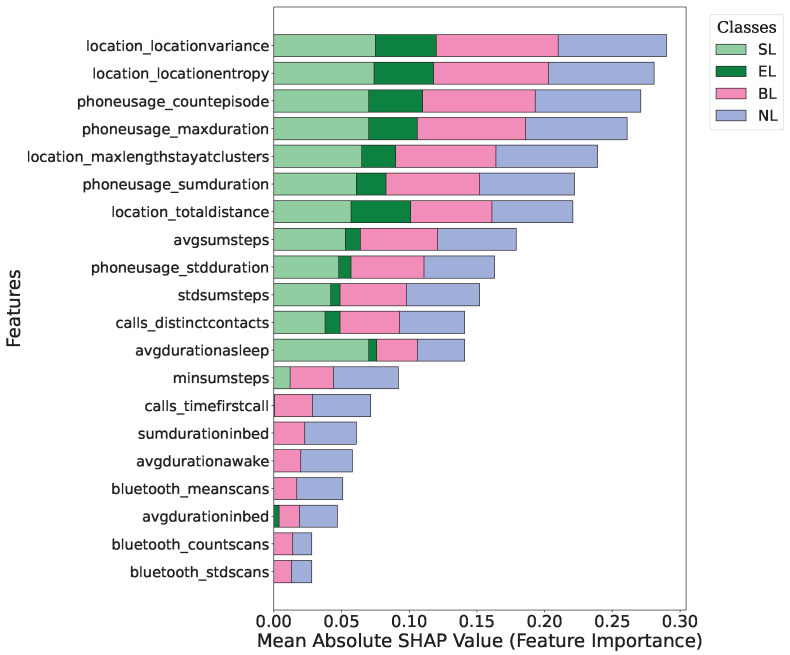
Mean Absolute SHAP values representing feature importance in the Random Forest model. Higher values indicate stronger contributions of a feature to the model’s decision-making process.

**Table 1 sensors-25-01903-t001:** Study Information and Participant Demographics. Gender acronyms: F: Female, M: Male, NB: Non-binary. Racial acronyms: A: Asian, B: Black or African American, H: Hispanic, N: American Indian/Alaska Native, PI: Pacific Islander, W: White, NA: Did not report. The & symbol denotes participants who identified with multiple races.

Category	Data
Participants	Total: 218
	Gender: F 111, M 107
	Ethnicity: A 102, B 6, H 10, N 2, PI 1, W 70,
	A&B 1, A&W 16, H&W 2, B&W 2,
	A&H&W 1, B&H&W 1, H&N&W 1, NA 3
Ground Truth	Pre-study 10-items UCLA scale
	Post-study 10-items UCLA scale
Sensor Data	Bluetooth, Wi-Fi, Call Logs, Location, Phone Usage, Physical Activity, Sleep

**Table 2 sensors-25-01903-t002:** Summary of sensor-derived features.

Category	Derived Sensed-Behaviours
Wi-Fi	Number of scans, unique devices, and scans of the most connected device measured over different time segments (day, morning, afternoon, evening, night).
Bluetooth	Number of scans, unique devices, mean and standard deviation of scans per device; most and least frequent device scans within segments, across segments and across the entire dataset.
Location	Total distance traveled, radius of gyration, maximum diameter, maximum distance from home, significant locations visited, flight lengths and durations (mean and standard deviation), fraction of day in a pause, Shannon entropy, circadian routine, weekday/weekend routine comparison, location variance, total distance using haversine formula, average and variance of speed during movement, transitions between locations, time spent at top locations, moving to static ratio, time in non-significant clusters, time spent at specific places.
Phone Usage	Total, longest, shortest, average, and standard deviation of unlock duration, number of unlock episodes, time until first unlock, and these metrics for specific places (living, exercise, study, greens).
Call	Number of calls, distinct contacts, mean, sum, minimum, maximum, standard deviation, mode, and entropy of call durations, time of first and last call, calls with the most frequent contact.
Physical Activity	Maximum, minimum, average, median, and standard deviation of daily steps, total steps; number and duration (total, max, min, average, standard deviation) of sedentary and active bouts.
Sleep	Number, total, longest, shortest, average, median, and standard deviation of awake/asleep episodes (main, nap, all), first and last wake/bedtimes, sleep efficiency, total and average duration to fall asleep, asleep, awake, in bed, and after wakeup for each sleep type.

**Table 3 sensors-25-01903-t003:** Division of 10-item UCLA scale into emotional and social loneliness; ’R’ indicates reverse scoring.

Type	Questions
Emotional Loneliness	1. How often do you feel that no one really knows you well?
	2. How often do you feel close to other people? (R)
	3. How often do you feel that there are people who really understand you? (R)
	4. How often do you feel that there are people you can turn to? (R)
	5. How often do you feel that people are around you but not with you?
Social Loneliness	1. How often do you feel that you have a lot in common with the people around you? (R)
	2. How often do you feel that you feel left out?
	3. How often do you feel isolated from others?
	4. How often do you feel that there are people you can talk to? (R)
	5. How often do you feel that you lack companionship?

**Table 4 sensors-25-01903-t004:** Summary of significant features between the SL and EL groups, including mean differences and effect sizes.

Features	SL Group	EL Group	MDiff	Effect Size (Cohen’s *d*)
**Location**				
LogLocationVariance(evening)	2.301 (1.864, 2.875)	3.751 (3.284, 4.324)	−1.452	−0.715 (−0.964, −0.514)
NumberOfSignificantPlaces	1.504 (1.67, 2.53)	2.167 (1.675, 2.602)	−0.663	−0.327 (−0.529, −0.264)
NumberOfLocationTransitions	5.463 (4.174, 6.732)	7.374 (5.638, 9.532)	−1.911	−0.780 (−0.957, −0.604)
NormalizedLocationEntropy	0.451 (0.403, 0.546)	0.323 (0.253, 0.350)	0.128	0.640 (0.549, 0.732)
**Phone Usage**				
SumDuration	400.204 (384.163, 416.432)	495.535 (480.862, 510.303)	−95.331	−0.535 [−0.758, −0.313]
CountEpisode	30.104 (25.562, 35.942)	40.645 (35.074, 45.521)	−10.541	−0.647 [−0.855, −0.439]
FirstUseAfter	28.745 [24.389, 33.867]	45.067 [41.483, 48.965]	−16.322	−0.578 [−0.739, −0.418]
MaxDuration	7.683 (6.422, 8.955)	18.073 (17.534, 19.131)	−10.390	−0.756 [−0.910, −0.603]
**Bluetooth**				
UniqueDevices	3.701 (2.485, 4.933)	5.516 (4.433, 6.597)	−1.815	−0.238 (−0.302, −0.175)
CountScans	13.231 (11.630, 14.842)	19.096 (17.751, 20.448)	−5.865	−0.277 (−0.452, −0.102)
**Steps**				
MaxSumSteps	5800.553 (5400.705, 6200.321)	6300.878 (5900.205, 6700.426)	−500.325	−0.518 (−0.604, −0.433)
AvgSumSteps	4800.335 (4500.634, 5100.074)	5300.745 (5000.832, 5600.642)	−500.410	−0.237 (−0.374, −0.128)
**Sleep**				
AvgDurationAwake	60.320 (48.294, 72.041)	88.385 (78.037, 102.181)	−28.065	−0.451 (−0.647, −0.255)
AvgDurationaSleep	510.047 (405.378, 610.582)	407.731 (385.284, 433.539)	102.316	0.404 (0.308, 0.501)

**Table 5 sensors-25-01903-t005:** Classification results for different models on loneliness categories.

Model	Acc	Socially Lonely			Emotionally Lonely			Both Lonely			Not Lonely		
		Prec	Rec	F1	Prec	Sens	F1	Prec	Rec	F1	Prec	Sens	F1
BL1: MC	42.54	0.00	0.00	0.00	0.00	0.00	0.00	42.54	100.00	59.59	0.00	0.00	0.00
BL2: DT	45.49	20.45	25.87	22.84	18.38	21.44	19.18	55.46	65.32	60.50	35.20	38.39	36.54
BL3: RWC	35.47	11.65	24.58	15.24	9.18	19.43	12.54	40.94	85.34	54.65	37.28	38.22	37.59
SVM	70.10	60.00	50.55	55.74	65.28	60.74	62.63	75.18	80.11	77.67	70.39	65.54	67.10
RF	75.58	65.43	55.74	60.54	70.15	65.24	67.08	80.84	85.72	82.41	75.94	70.86	72.76
KNN	65.53	55.11	45.69	50.64	60.48	55.00	57.06	70.15	75.75	72.38	65.57	60.92	62.14
XGBoost	78.48	68.37	58.59	63.62	68.63	73.27	70.49	88.07	83.13	85.44	73.58	78.52	75.27

## Data Availability

The data used in this study, the GLOBEM dataset, are available at https://the-globem.github.io/ (accessed on 17 May 2024).

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
