# Peer review of "Unmasking Nuances Affecting Loneliness: Using Digital Behavioural Markers to Understand Social and Emotional Loneliness in College Students"

_sensors, 2025, doi:10.3390/s25061903_

Round 1
Reviewer 1 Report
Comments and Suggestions for Authors
This study proposed a method utilizing passively sensed data to distinguish between social and emotional loneliness among college students. The work demonstrates both practical importance and novelty. The manuscript is generally well-written, presenting a clear and logically structured narrative. However, several areas require more refinement, as outlined below:
- The introduction is overly verbose. It is recommended to merge the first and second paragraphs to improve conciseness. Additionally, the literature review should be strengthened by explicitly addressing the current state of research on loneliness detection methodologies, particularly those employing passive sensing technologies.
- Some questions about algorithm citations:
On page 6, Line 180: Mutual Information (MI) is an established algorithm; a source citation is required here.
On page 7, Line 225: Similarly, references are needed for SVM, XGBoost, Random Forest, and KNN.
A thorough review of the manuscript is advised to ensure proper attribution for all established methodologies.
3.On Page 7, Line 214, the use of "Cohen's d" requires clarification. If this metric is not original to the authors, please cite the original source (Cohen, 1988, or later relevant work) ,otherwise,justify its selection over alternative effect size measures in the context of this study.
4.On page 9, Lines 311-319: Inconsistent use of units (e.g., "M" values followed by "minutes" in some cases but not others) should be addressed. A brief explanation for this discrepancy is necessary.
- The horizontal axes lack clear labels in Figure 2 and 3. Define the meaning of the coordinate values (e.g., time intervals, feature rankings) in these two figures. Meanwhile, font styles in these figures deviate from the main text. Standardize typography to match manuscript guidelines.
6.The reference list includes predominantly older sources. Incorporate recent publications (e.g., 2024 or 2025) to reflect advancements in passive sensing and loneliness research.
Recommendation
While this study holds significant potential, I recommend Major Revision to address the above concerns. The authors should provide a point-by-point response to all critiques and implement corresponding revisions.
Suggest the author to verify, check and improve the English expression of the manuscript again.
Author Response
Comment 1: The introduction is overly verbose. It is recommended to merge the first and second paragraphs to improve conciseness. Additionally, the literature review should be strengthened by explicitly addressing the current state of research on loneliness detection methodologies, particularly those employing passive sensing technologies.
Response 1: Thanks for the comment. We have now merged the first and second paragraphs of the introduction. The revised version reduces redundancies and improves readability. To strengthen the literature review on loneliness detection methodologies, we have explicitly added recent research on passive sensing techniques.
Comment 2: Some questions about algorithm citations:
On page 6, Line 180: Mutual Information (MI) is an established algorithm; a source citation is required here.
On page 7, Line 225: Similarly, references are needed for SVM, XGBoost, Random Forest, and KNN.
A thorough review of the manuscript is advised to ensure proper attribution for all established methodologies.
Response 2: Thanks for the comment. The references has been added.
Comment 3: On Page 7, Line 214, the use of "Cohen's d" requires clarification. If this metric is not original to the authors, please cite the original source (Cohen, 1988, or later relevant work) ,otherwise,justify its selection over alternative effect size measures in the context of this study.
Response 3: Thanks for the comment. We have added the original reference of Cohen’s d. Also, we included the justification for using this. Changes have been highlighted in blue.
Comment 4: On page 9, Lines 311-319: Inconsistent use of units (e.g., "M" values followed by "minutes" in some cases but not others) should be addressed. A brief explanation for this discrepancy is necessary.
Response 4: Thanks for the comment. We have now explicitly stated units for all reported metrics to ensure clarity.
Comment 5: The horizontal axes lack clear labels in Figure 2 and 3. Define the meaning of the coordinate values (e.g., time intervals, feature rankings) in these two figures. Meanwhile, font styles in these figures deviate from the main text. Standardize typography to match manuscript guidelines.
Response 5: Thanks for the comment. We have added clear axis labels in Figures 2 and 3 (which are now Fig 4 and 5). Regarding font style, we have ensured consistency by using Times New Roman which is a widely accepted and commonly used font and this aligns with MDPI’s recommendation to use standard fonts in figures (https://www.mdpi.com/authors/layout)
Comment 6: The reference list includes predominantly older sources. Incorporate recent publications (e.g., 2024 or 2025) to reflect advancements in passive sensing and loneliness research.
Response 6: Thanks for the comment. We have added recent publications references in the Introduction section’s literature review paragraphs to ensure our work reflects the latest advancements in passive sensing and loneliness research.
Reviewer 2 Report
Comments and Suggestions for Authors
In this paper, the authors investigate behavioral patterns associated with social and emotional loneliness using passively sensed data from a student population. Using statistical analysis, machine learning, and SHAP-based feature importance methods, the result shows that there are significant differences in behaviors between socially and emotionally lonely students. Specifically, there were distinct differences in phone use and location-based features. The result analysis shows a strong ability to classify types of loneliness accurately. The XGBoost model achieved the highest accuracy (78.48%) in predicting loneliness. The research is interesting. However, the following issues should be clarified.
- In the introduction part, the authors did not cite and analyze the research effect related to the social and emotional loneliness based on statistical analysis, machine learning. Please revise.
- In the Method part. Please provide the flowchart of the proposed method.
- In the section of Results, some necessary figures should be provided. For example, the confusion matrix of the different methods for classifying types of loneliness could be provided.
- Why the XGBoost show better performance in predicting loneliness?
- In Table.2. The data are collected from different type of sensors. Also, the dimension of sensor-derived features is very high. How to determine the important features which are crucial for loneliness predication?
The Quality of English Language sounds good.
Author Response
Comment 1: In the introduction part, the authors did not cite and analyze the research effect related to the social and emotional loneliness based on statistical analysis, machine learning. Please revise.
Response 1: Thanks for the comment. We have added relevant citations in the Introduction’s related work part to discuss studies that have used statistical analyses to analyse social and emotional loneliness. Specifically, we highlight research on elderly populations and community studies which show distinct behavioral and psychological patterns associated with each loneliness type. We also emphasized the research gap in differentiating social and emotional loneliness in younger populations through behavioural data and machine learning methods.
Comment 2: In the Method part. Please provide the flowchart of the proposed method.
Response 2: Thanks for the comment. A flow chart (Figure. 1) has been added in methods.
Comment 3: In the section of Results, some necessary figures should be provided. For example, the confusion matrix of the different methods for classifying types of loneliness could be provided.
Response 3: Thanks for the comment. We have added the confusion matrix (Figure. 3) for XGBoost model, since it has shown the best performance among all models.
Comment 4: Why the XGBoost show better performance in predicting loneliness?
Response 4: Thanks for the comment. In the discussion section, we have discussed some reasons on why XGBoost could have potentially shown best performance including its ability to capture non linear relationships within data. The changes have been highlighted in blue.
Comment 5: In Table.2. The data are collected from different type of sensors. Also, the dimension of sensor-derived features is very high. How to determine the important features which are crucial for loneliness predication?
Response 5: Thanks for the comment. To determine the most crucial sensor derived features for loneliness prediction, we performed feature importance analysis using SHAP. The results section 3.4 (Important Features for Loneliness Classification) presents the top contributing features to identify behavioural patterns strongly associated with loneliness.
Round 2
Reviewer 2 Report
Comments and Suggestions for Authors
The revised paper is acceptable for publication.